# The relationship between gender discrimination and wellbeing in middle-aged and older women

**Ruth A. Hackett**[1,2]*, **Myra S. Hunter**[1], **Sarah E. Jackson**[2]

**1** Institute of Psychiatry, Psychology and Neuroscience, Health Psychology Section, King's College London, London, United Kingdom, **2** Department of Behavioural Science and Health, University College London, London, United Kingdom

* ruth.hackett@kcl.ac.uk

## Abstract

### Background

Emerging evidence suggests that perceived gender discrimination negatively impacts mental wellbeing in young women.

### Purpose

This study explored whether a similar relationship exists in middle-aged and older women.

### Methods

A total of 3081 women (aged ≥52 years) from the English Longitudinal Study of Ageing provided data on perceived gender discrimination in 2010/11. Depressive symptoms, loneliness, quality of life and life satisfaction were assessed in 2010/11 and in 2016/17.

### Results

Perceived gender discrimination was reported by 282 (9.2%) participants. Cross-sectionally, women who perceived gender discrimination reported more depressive symptoms (β = 0.34, 95% CI 0.11 to 0.57) and had higher loneliness scores (β = 0.14, 95% CI 0.08 to 0.20) than women who did not perceive gender discrimination. They also reported significantly lower quality of life (β = −2.50, 95% CI −3.49 to −1.51) and life satisfaction (β = −1.07, 95% CI −1.81 to −0.33). Prospectively, perceived gender discrimination was associated with greater loneliness scores (β = 0.08, 95% CI 0.02 to 0.14), as well as lower ratings of quality of life (β = −0.98, 95% CI −0.09 to −1.86), and life satisfaction (β = −1.04, 95% CI −0.34 to −-1.74), independent of baseline values.

### Conclusions

Middle-aged and older women who perceive gender discrimination report poorer mental wellbeing than those who do not perceive discrimination. Further, this type of discrimination may be predictive of declining mental wellbeing over time. These findings highlight the need

**Data Availability Statement:** De-identified data from this study and all materials used to conduct the study are available in public archives: https://g2aging.org and https://www.elsa-project.ac.uk/.

**Funding:** This study was supported by the Academy of Medical Sciences/the Wellcome Trust/ the Government Department of Business, Energy and Industrial Strategy/the British Heart Foundation/Diabetes UK Springboard Award [SBF006\1036]. This funding was awarded to RAH. The funders had no role in the study design, data collection and analysis, decision to publish, or preparation of the manuscript. https://acmedsci.ac. uk/.

**Competing interests:** The authors have declared that no competing interests exist.

for interventions to target gender-based discrimination to improve the wellbeing of women at mid- and older age.

## Introduction

Globally, populations are ageing [1,2]. This phenomenon represents a public health challenge [3,4], as ageing is associated with a decline in physical and mental capacity, increasing the risk of poor health and wellbeing [3,5]. It is acknowledged that these age-related changes in health and mental wellbeing are not equally distributed across the population [3,6]. For example, accumulating evidence suggests mental health disorders such as depression are more prevalent in women than men [7,8], and older women typically report more depressive symptoms and poorer wellbeing than their male counterparts [9]. In addition, the impact of mental disorders and poor wellbeing on indicators of healthy ageing is suggested to be greater in women than in men [10].

Several explanations have been offered for these sex patterns in mental wellbeing, including biological differences (e.g., variation in hormone levels or the reproductive risk of postnatal mental disorders) and women's greater risk of experiencing interpersonal stressors (e.g., sexual violence) [8,11]. Another possibility is that perceptions of discrimination attributed to gender may negatively influence women's mental wellbeing.

Discrimination is defined as the differential treatment of a person based on a socially ascribed characteristic such as gender [12]. Discrimination based on gender is perceived to be common, with 37% of European adults reporting it to be widespread, based on a survey of almost 28,000 participants [13]. Perceived gender discrimination may operate at both the individual and structural level to impact wellbeing. For some women, perceptions of gender discrimination may occur at the individual level through negative day-to-day interactions and experiences with others. Reviews in the area have focused on the negative impact of workplace gender discrimination and sexual harassment [14,15], though gender -based discrimination has also been described on the street [16,17] and on public transport [18]. For other women, perceived gender discrimination may operate at the structural level. Such structural inequalities are reflected in the fact that globally women are less likely to hold positions of economic, social or political power than men [19], despite legislative efforts to increase sex equality [20]. For middle-aged and older women in the United Kingdom (UK), the right to equal pay [21] and the recognition of sex as protected characteristic under equality law [22] was secured during their lifetimes. However, this legal equality has not fully translated into equal experiences as a 'gender pay gap' still exists [23] and gender-based discrimination is perceived to be common [13,24].

An emerging body of work has investigated the impact of perceived discrimination on mental wellbeing [25–27]. Systematic reviews and meta-analyses have linked perceived discrimination with depression, psychological distress and poor life satisfaction among other outcomes [25–27]. The largest review to date included 328 studies and of these 23 focused on gender discrimination [27]. In an independent analysis, perceived gender discrimination was associated with poor mental wellbeing [27]. However, all these studies were cross-sectional in nature, meaning the temporal order of associations could not be determined.

Two subsequent studies have addressed this gap in the literature by assessing prospective, as well as cross-sectional associations between perceived gender discrimination and mental wellbeing. In a UK-based study, perceived gender discrimination was linked with poor mental

wellbeing as in previous work [28]. In this study of almost 3000 young women, 19.5% of the sample perceived gender discrimination. Those who perceived gender discrimination had greater psychological distress, worse mental functioning and lower life satisfaction than those who did not perceive gender discrimination over 4-year follow-up [28]. These findings were independent of wellbeing at the time of the discrimination assessment.

One other study has investigated both cross-sectional and prospective associations between perceived gender discrimination and mental wellbeing. In a US-based study of over 6000 middle-aged and older men and women, perceived gender discrimination was linked with increased loneliness in cross-sectional analyses [29]. However, no prospective associations between perceived gender discrimination and changes in loneliness at 4-year follow-up were observed. In addition, no significant cross-sectional or prospective associations with life satisfaction were detected in this sample.

It is unclear why these studies differed in their findings, as both benefitted from large sample sizes and a four-year follow-up period. One possibility is that age differences accounted for the diverging findings, as the UK-sample were considerably younger on average (mean age 38.68 years) [28] than the US-sample (mean age 67 years) [29]. Differences in the prevalence of perceived gender discrimination may also have played a role, as rates were higher in the UK compared with the US sample (19.5% vs 13%), though it is likely these higher rates are at least partially accounted for by the fact that the UK study assessed women alone, while the US study also included men.

Taken together, few studies have assessed prospective associations between perceived gender discrimination and mental wellbeing. This previous work has produced conflicting findings, with significant associations observed in younger UK-based women and no such associations observed in middle-aged and older people in the US. It is unclear whether the mixed findings are accounted for by country- or age-related differences. To help clarify these issues, the current study aimed to investigate cross-sectional and prospective associations between perceived gender discrimination and mental wellbeing in a community-dwelling sample of middle-aged and older women living in England. Specifically, we assessed possible associations with measures of depressive symptoms, loneliness, quality of life and life satisfaction.

## Method and materials

### Study data and sample

We used data from the English Longitudinal Study of Ageing (ELSA), a prospective cohort of people aged 50 years and older living in England [30]. Data collection started in 2002 (wave 1) with follow-up waves taking place biennially. ELSA data collection is ongoing. Every wave data collection happens through self-completion questionnaires and computer-assisted personal interview. In alternate waves anthropometric data are obtained. Here we use baseline data from wave 5 (2010–11; the only time point in which discrimination was measured) and follow-up data from wave 8 (2016–17; as in other ELSA studies on discrimination [31–35]). Data was freely available to download from the UK Data Service.

We restricted our analyses to women. In wave 5 of ELSA there 5,705 were women. Of these 4,095 provided information on demographic characteristics, health behaviours and perceived discrimination. We removed 1,013 participants with missing data on body mass index (BMI). BMI was objectively measured in ELSA and was included in our statistical models given previous work associating weight with both discrimination and wellbeing [36]. After these exclusions we had a final sample size of 3,081 women. ELSA received ethical approval from the

London Multicentre Research and Ethics Committee (MREC/01/02/91). All participants gave full written informed consent.

## Measures

### Perceived discrimination

The women in the current study were asked about the frequency of encounters with five discriminatory situations: *"In your day-to-day life, how often have any of the following things happened to you: 1) you are treated with less respect or courtesy; 2) you receive poorer service than other people in restaurants and stores; 3) people act as if they think you are not clever; 4) you are threatened or harassed; and 5) you receive poorer service or treatment than other people from doctors or hospitals."* Response options were on a 6-point scale ranging from 'never' to 'almost every day'. As the data were skewed, with most women 'never' reporting discrimination, we dichotomised responses to indicate whether or not they perceived discrimination in the past year (a few times or more a year vs less than once a year or never), with the exception of the fifth item which was dichotomised to indicate whether or not respondents had ever experienced discrimination from doctors or hospitals (never vs all other options) as most individuals never reported discrimination in this setting. In line with previous work in ELSA, responses were combined to create an overall discrimination binary score (yes/no) if participants reported any of these experiences [24,31–35]. If participants reported discrimination in any of the situations, a follow-up question asked participants to indicate the characteristic(s) they attributed their experience to, with a choice from a list of options including age, race, sex, sexual orientation, and weight. Respondents could choose more than one option from the list. In the current study, women who attributed the discriminatory experience to their sex were categorised as cases of perceived gender discrimination. These discrimination items have been used widely to assess associations between discrimination and wellbeing in ELSA [31–34] and other longitudinal studies [37,38].

### Wellbeing measures

The Center for Epidemiologic Studies Depression Scale (CES-D) was used to measure depressive symptoms [39]. This 8-item scale included statements such as *'I felt sad* and *'I could not get going'* rated over the past month with response options of yes/no. The overall score ranged from 0–8, with higher values indicating greater symptomatology [39].

We used the Revised University of California, Los Angeles (UCLA) Loneliness Scale [40] was used to assess loneliness. This 3-item scale included questions such as *'How often do you feel left out?'* with response options of 1, *'hardly ever/never'*; 2, *'some of the time'*; and 3, *'often'*. The responses were averaged to produce an overall score. This ranged from 0–3, with higher values indicating greater loneliness [41].

We used the CASP-19 [42] to assess quality of life. This scale measures quality of life at older age. The 19-items cover different aspects of quality of life including autonomy, control, pleasure and self-realisation. Participants were asked how often each item applies to them with response options from 0 *'often'* to 3 *'never'*. The overall score ranged from 0–57, with higher values indicating higher quality of life.

The Satisfaction With Life Scale [43] was used to measure life satisfaction. Respondents rated the extent of their agreement with five items (e.g., *"The conditions of my life are excellent"*) with response options from 0 (*strongly disagree*) to 6 (*strongly agree*). Responses were summed producing an overall score (range: 0–30). Higher scores indicate more life satisfaction.

## Covariates

We selected our covariates in advance due to associations with discrimination and wellbeing reported in earlier research [24,31,32,44,45]. All covariates (except BMI) were assessed at baseline (wave 5, 2010–11) and were self-reported. Age was measured in years and ethnicity was coded as (white/ethnic minority). Marital status was coded as married vs. single/separated/ divorced/widowed). We adjusted for household non-pension wealth (reported in quintiles) as this is considered the best measure of socio-economic position in the ELSA cohort [30]. Smoking was coded as non-smoker/smoker. The frequency of *"vigorous/moderate/mildly energetic"* physical activity was coded in binary (non-sedentary = *"more than once a week/once a week/ one to three times a month"* vs sedentary = *"hardly ever or never")*. BMI was not assessed as at wave 5. Therefore, objectively measured height (cm) and weight (kg) data from wave 4 (2008–09) were used to derive BMI (kg/m$^2$).

## Statistical analysis

We compared the descriptive characteristics of women who did and did not perceive gender discrimination using $\chi^2$ tests for categorical variables and independent-samples *t*-tests for continuous variables at baseline (wave 5, 2010–2011).

Differences in depressive symptoms, loneliness, quality of life and life satisfaction between those who perceived gender discrimination and those who did not were assessed in both cross-sectional and prospective analyses. Linear regression models adjusted for age, ethnicity, marital status, wealth, smoking, physical activity, and BMI. Prospective analyses over 6-year follow-up additionally controlled for baseline status/score on the wellbeing measure of interest. We present the results as unstandardized B and 95% confidence intervals (CI).

We tested for interactions between gender discrimination and age and ethnicity on wellbeing outcomes in preliminary analyses. No significant moderation was observed, so interaction terms are not included in the final models. We also investigated whether including women who were missing BMI data in our sample changed our findings. As the results were similar (see S1 Table), we restricted our analyses to participants with complete information on BMI.

We carried out several sensitivity analyses. Firstly, we assessed whether participants who were lost to follow-up (*n* = 789) differed from those who provided both cross-sectional and prospective data (*n* = 2,292). We then assessed whether this influenced the findings by carrying out the cross-sectional analyses (wave 5) including only participants who had follow-up data at wave 8. For our second sensitivity analysis, we assessed whether the prospective findings from our main analysis (complete case analysis at wave 8) were similar when we imputed the missing outcome information (using multiple imputation with baseline outcome information and covariates as predictors) for those respondents lost to follow-up (*n* = 789). We created 20 imputed datasets and analysed each separately. Then the findings were combined to produce pooled estimates of effects. These pooled effects are reported for the sensitivity analysis. For our third sensitivity analysis, we tested the possibility that one of the five discriminatory experiences contributing to the measure of perceived gender discrimination was driving the findings. We assessed this this by repeating our cross-sectional and prospective analyses removing each type of discriminatory experience in turn. Analyses were not pre-registered and were conducted using SPSS version 26.

## Results

### Participant characteristics

Perceived gender discrimination was reported by 282 (9.2%) participants (Table 1). The most common discriminatory experiences reported by these women were *"being treated with less*

**Table 1. Characteristics of women by perceived discrimination at wave 5 of the English Longitudinal Study of Ageing (2010/11).**

| | | No perceived discrimination (*n* = 2799) | Perceived discrimination (*n* = 282) | *p* |
|---|---|---|---|---|
| Age (years) | | 67.66 (8.75) | 64.39 (7.25) | < 0.001 |
| Ethnicity (% white) | | 2753 (98.4%) | 273 (96.8%) | = 0.061 |
| Marital status (% married) | | 1665 (59.5%) | 172 (61.0%) | = 0.623 |
| Wealth quintile (£) | | | | = 0.045 |
| | 1 | 477 (17.0%) | 35 (12.4%) | |
| | 2 | 582 (20.8%) | 52 (18.4%) | |
| | 3 | 562 (20.1%) | 53 (18.8%) | |
| | 4 | 564 (20.2%) | 75 (26.6%) | |
| | 5 | 614 (21.9%) | 67 (23.8%) | |
| Body Mass Index (kg/m$^2$) | | 28.21 (5.56) | 28.59 (5.65) | = 0.270 |
| Smoking (% yes) | | 316 (11.3%) | 32 (11.3%) | = 0.977 |
| Physical activity (% sedentary) | | 481 (17.2%) | 24 (8.5%) | < 0.001 |
| Perceived discrimination (% yes) | | | | |
| Treated with less respect/courtesy | | - | 232 (82.3%) | - |
| Poorer service in restaurants/stores | | - | 124 (44.0%) | - |
| People act as if you are not clever | | - | 140 (49.6%) | - |
| Threatened or harassed | | - | 49 (17.4%) | - |
| Poorer service from doctors/ hospitals | | - | 32 (11.4%) | - |

Data are presented as means (SD) and n (%).

*respect or courtesy"* (82.3%), *"people acting as if you are not clever"* (49.6%), and *"receiving poorer service than other people in restaurants and stores"* (44.0%). Being *"threatened or harassed"* (17.4%) or *"receiving poorer service or treatment than other people from doctors or hospitals"* (14.6%) were less frequently reported.

The average age in the sample was 67.46 years (standard deviation = 8.67) and 1.8% (n = 55) of the sample reported being from an ethnic minority group. In comparison to women who did not perceive gender discrimination, those who perceived discrimination were significantly younger (67.66 ± 8.75 vs 64.39 ± 7.25 years, *p* < 0.001) and wealthier (highest wealth quintile 21.9% vs 23.8%, *p* = 0.045) on average (Table 1). They were also less likely to be sedentary (17.2% vs 8.5%, *p* < 0.001) than those who did not perceive gender discrimination. The groups did not significantly differ in terms of ethnicity, marital status, BMI or smoking behaviour (*p* > 0.061).

## Cross-sectional associations between perceived gender discrimination and wellbeing

Cross-sectionally, women who perceived gender discrimination reported a greater number of depressive symptoms on average (β = 0.34, 95% CI 0.11 to 0.57) than women who did not

**Table 2. Cross-sectional and prospective associations between perceived discrimination and health and wellbeing outcomes (complete cases).**

| | | | Wave 5 (cross-sectional) | | | | Wave 8 (prospective) | |
|---|---|---|---|---|---|---|---|---|
| | | n | No perceived discrimination | n | Perceived discrimination | n | No perceived discrimination | n | Perceived discrimination |
| Depression | | | | | | | | | |
| | Mean score (SE) | 2765 | 1.55 (0.04) | 281 | 1.89 (0.11) | 2046 | 1.48 (0.04) | 217 | 1.56 (0.11) |
| | Coeff. [95% CI] | | Ref | | 0.34 [0.11; 0.57]** | | Ref | | 0.08 [-0.16; 0.31] |
| Loneliness | | | | | | | | | |
| | Mean score (SE) | 2775 | 1.41 (0.01) | 280 | 1.55 (0.03) | 1875 | 1.36 (0.01) | 202 | 1.44 (0.03) |
| | Coeff. [95% CI] | | Ref | | 0.14 [0.08; 0.20]*** | | Ref | | 0.08 [0.02; 0.14]* |
| Quality of life | | | | | | | | | |
| | Mean score (SE) | 2678 | 41.53 (0.15) | 273 | 39.03 (0.48) | 1722 | 42.33 (0.14) | 194 | 41.36 (0.43) |
| | Coeff. [95% CI] | | Ref | | -2.50 [-1.51; -3.49]*** | | Ref | | -0.98 [-0.09; -1.86]* |
| Life satisfaction | | | | | | | | | |
| | Mean score (SE) | 2691 | 20.65 (0.12) | 278 | 19.58 (0.36) | 1774 | 21.01 (0.11) | 199 | 19.97 (0.34) |
| | Coeff. [95% CI] | | Ref | | -1.07 [-1.81; -0.33]** | | Ref | | -1.04 [-0.34; -1.74]** |

All analyses are adjusted for age, wealth, ethnicity, marital status, body mass index, smoking and physical activity. Prospective analyses are additionally adjusted for baseline scores/status.

Coeff = unstandardized B coefficient, CI = confidence interval.

*$p < 0.05$

**$p < 0.01$

***$p < 0.001$.

Possible scores on the depression measure range from 0–8, on the loneliness measure range from 1–3, on the quality of life scale range from 0–57, and on the life satisfaction scale range from 0–30.

perceive gender discrimination (Table 2; first panel). This association was independent of age, wealth, ethnicity, marital status, BMI, smoking and physical activity. Perceived gender discrimination was also significantly associated with higher loneliness scores (β = 0.14, 95% CI 0.08 to 0.20), independent of covariates. Additional adjustment for depressive symptoms in the loneliness analyses did not change the pattern of results (β = 0.10, 95% CI 0.05 to 0.16, $p < 0.001$). Those who perceived gender discrimination also had significantly lower quality of life (β = −2.50, 95% CI −3.49 to −1.51) and life satisfaction (β = −1.07, 95% CI −1.81 to −0.33) than those who did not perceive gender discrimination.

## Prospective associations between perceived gender discrimination and wellbeing

Prospectively, perceived gender discrimination was associated with higher loneliness scores (β = 0.08, 95% CI 0.02 to 0.14), independent of covariates and baseline loneliness scores (Table 2; second panel). The inclusion of baseline depressive symptoms in the loneliness model did not change the pattern of results (β = 0.07, 95% CI 0.02 to 0.13). Women who perceived gender discrimination also had lower ratings of quality of life (β = −0.98, 95% CI −0.09 to −1.86), and

life satisfaction (β = −1.04, 95% CI −0.34 to −1.74), independent of covariates and baseline values. No significant prospective association between perceived gender discrimination and depressive symptoms was detected.

## Sensitivity analyses

We investigated whether participants who were lost to follow-up ($n$ = 789) differed from those who had data at waves 5 and 8 ($n$ = 2,292). The findings of this first sensitivity analysis can be found in S2 Table. Those lost to follow-up were older on average (70.89 ± 10.06 vs 66.15 ± 7.78 years, $p < 0.001$) and were less likely to be married (52.9% vs 62%, $p < 0.001$) than those who had full data. They were also less wealthy ($p < 0.001$) and were more likely to be smokers (14.2% vs 10.3%, $p = 0.004$) and to be sedentary (29.0% vs 12.0%, $p < 0.001$). We assessed whether these differences influenced our findings by carrying out our cross-sectional analyses (wave 5) including only those who had follow-up data (wave 8; $n$ = 2,292). The pattern of results remained unchanged (S3 Table). In the second sensitivity analysis, we assessed whether the prospective findings from our main (complete case) analysis were similar when missing outcome data was imputed for respondents who were lost to follow-up ($n$ = 789). This did not alter the findings (S4 Table). In the final sensitivity analysis, we assessed whether one of the five types of discriminatory experience contributing to the measure of perceived gender discrimination was driving the findings. Our cross-sectional findings were unchanged (S5 Table, upper panel). Similarly, the prospective findings were mostly unchanged except when removing "*being treated with less respect*" from the measure of perceived gender discrimination the association between gender discrimination and quality of life was attenuated (β = −0.75, 95% CI −1.58 to 0.09, $p = 0.080$).

## Discussion

In this study, we assessed associations between perceived gender discrimination and wellbeing in a large prospective sample of middle-aged and older women living England. In cross-sectional analyses, women who perceived that they had experienced gender discrimination in their everyday lives reported more depressive symptoms, being lonelier, having poorer quality of life, and being less satisfied with their lives than those who did not perceive gender discrimination. These results held prospectively for loneliness, quality of life, and life satisfaction, controlling for covariates including baseline scores on these measures. No significant prospective association with depression was observed.

A limited number of previous studies have assessed the impact of perceived gender discrimination on mental wellbeing over time. The results of the current study add to the literature by demonstrating a prospective relationship between perceived gender discrimination and poor mental wellbeing over a 6-year follow-up period. These findings in a sample of middle-aged and older women in England are in agreement with results from an earlier study of younger women in the UK [28]. Taken together, these prospective findings indicate that perceptions of gender discrimination may be predictive of declining mental wellbeing over time in general population cohort samples in England and the UK. These findings also align with previous prospective work linking sexual harassment with poor mental wellbeing in student [46–48] and working samples [49].

However, not all previous studies have detected associations between perceived gender discrimination and changes in mental wellbeing [29]. In a US population cohort sample of middle-aged and older people perceived gender discrimination was not significantly associated with changes in loneliness or life satisfaction at four-year follow-up [29]. This contrasts with the results of the current study; whereby perceived gender discrimination was associated with

greater loneliness and poorer life satisfaction over 6-year follow-up, independent of baseline values on these measures. The explanation for these diverging findings is unclear, as both studies included adults of a similar age, had large sample sizes, multiyear follow-up periods and used the same measure of perceived gender discrimination. It is unlikely that differences in the prevalence of perceived gender discrimination played a role, as rates were higher in the US-based sample as reported in other work [24]. However, it is plausible that the current study offered greater precision in investigating the links between perceived gender discrimination and mental wellbeing by restricting the analyses to women. Thereby directly comparing outcomes in women who did and did not perceive gender discrimination.

In keeping with previous work, we detected significant cross-sectional relationships between perceived gender discrimination and poorer mental wellbeing [27,28]. Associations between perceived gender discrimination and depressive symptoms [28,50,51], loneliness [29] and poorer life satisfaction [28,51,52] have been reported previously. Our study adds to the cross-sectional literature by demonstrating these associations for the first time in a large population-based sample of middle-aged and older women living in England. Cross-sectional analyses cannot ascertain whether perceived gender discrimination leads to poor mental wellbeing or whether perceptions of gender discrimination are indicative of psychological distress. Our prospective findings help disentangle such issues by establishing that perceived gender discrimination predicts loneliness, poorer quality of life and life satisfaction at 6-year follow-up, independent of baseline associations. Thus, suggesting that perceived gender discrimination has negative consequences for future wellbeing. Depressive symptoms were associated with perceived gender discrimination in cross-sectional but not in prospective analyses. This null result may suggest that the effect of ongoing gender discrimination on depressive symptoms was apparent at the baseline assessment, thus limiting the scope for further significant deterioration.

Several potential pathways could underlie the link between perceived gender discrimination and poor mental wellbeing. It is possible that perceived discrimination could operate through poor health behaviour to negatively impact wellbeing. For instance, perceived gender discrimination could act as a barrier to a healthy lifestyle (e.g., women avoiding running outdoors to avoid street harassment). Alternatively, health behaviour could be used as a method of coping with the distress associated with perceived gender discrimination (e.g., comfort eating, or smoking). Indeed, previous work has associated perceived gender discrimination with smoking [53], binge drinking [48,53], hard drug use [54] and restless sleep [55]. However, our analyses were robust to adjustment for smoking behaviour, physical activity and BMI. Further work is needed to ascertain the role of other health behaviours in the relationship between perceived gender discrimination and mental wellbeing.

Another possible mechanism that could help explain our findings is disturbances in stress-related biology. Under the theory of allostatic load, repeated exposure to stressors such as perceived discrimination causes frequent activation of the stress response systems. Over time this can lead to 'wear and tear' resulting in disturbances in multiple biological systems [56]. Research on the link between perceived discrimination with stress-related biology is dominated by studies on racism [26,57,58]. Pooled evidence has associated perceived discrimination with heightened cardiovascular responses to standardised laboratory stress [26,58], though none of the included studies focused on gender discrimination. Outside of the laboratory environment, sexual harassment has been linked with raised systolic blood pressure in a study of over 1000 participants [59]. Another stress-related biological process that may be implicated in the discrimination-wellbeing link is activation of the hypothalamic-pituitary-adrenal (HPA) axis. Indeed, changes in cortisol output have been related with both race [57,58,60] and weight discrimination [61]. To date, no study has assessed the link between gender discrimination

and cortisol in a naturalistic setting. However, in the laboratory, exposure to sexist scenarios have been associated with cortisol reactivity [62,63]. Considering the limited evidence linking perceived gender discrimination with stress-related biology, more work is needed in this area.

Our study benefitted from the use of a large well characterised sample of middle-aged and older women. The longitudinal nature of the ELSA study allowed us to assess changes in wellbeing over time, adding to the limited prospective literature on perceived gender discrimination. We were able to adjust for potential confounders including sociodemographic and behavioural factors.

However, our study was not without weaknesses. Our measure of gender discrimination was self-reported and reflects perceptions of discrimination rather than objective exposure to discriminatory events. Subjective interpretations of gender discrimination compared with objective encounters with gender discrimination could have differing impacts on mental wellbeing [27]. Only 9.2% of our sample reported gender discrimination. However, there is evidence that the tendency to minimize or deny personal discrimination is prevalent among women [64]. This has an impact on mental wellbeing, with evidence from large cohort studies suggesting that denial of gender discrimination is linked with greater mental wellbeing [65,66]. This denial of discrimination is suggested to be motivated by a desire to see the world as fair (known as system-justifying beliefs) and this may be beneficial for mental wellbeing [65–67]. We were unable to investigate whether system-justifying beliefs influenced the reporting of gender discrimination and in turn the links between gender discrimination and mental wellbeing in this study due to a lack of data availability. This represents an important avenue for future work. Our measure of perceived discrimination was not specific to gender discrimination as participants could attribute multiple reasons for the discrimination. While this could have helped avoid priming or bias, other measures (e.g., the Schedule of Sexist Events [68]) with tailored items on experiences of sexism could have garnered different findings. Further research is needed on how perceived gender discrimination interacts with other forms of discrimination to impact on wellbeing. Discrimination was only measured at one timepoint in ELSA. Therefore, it was not possible to assess whether experiences of gender discrimination were ongoing or changed over time. However, related work suggests that reports of sexual harassment are strongly predictive of later reports of sexual harassment [46,49]. Our results demonstrated a temporal relationship between perceived gender discrimination and later declines in wellbeing. There is some evidence that individuals with poor mental wellbeing may be more likely to report discrimination [69] and that sexual harassment and mental health are bidirectionally related [48]. Research testing reciprocal relationships between perceived gender discrimination and wellbeing could shed light on these issues. As women are more likely to perceive gender discrimination, we restricted our analyses to women. Nevertheless, women are also more likely than men to report poor mental wellbeing [9]. Few men in ELSA perceived gender discrimination. Therefore, we were underpowered to assess any possible associations. Our sample was largely of white ethnicity. Therefore, our findings may not generalise to ethnic minority groups.

Overall, this study adds to the literature by demonstrating that middle-aged and older women who perceive gender discrimination may experience declines in mental wellbeing. These findings highlight the need to reduce sexism, to promote equality and plausibly benefit mental wellbeing too. Indeed, there is evidence that women living in more gender equal societies have better mental wellbeing [70–72]. Perceptions of discrimination can act as a springboard for building collective movements to bring about social change. Interestingly, there is some evidence from small studies that speaking out online about sexism may also enhance wellbeing [73,74]. More research on the pathways underlying the gender discrimination-wellbeing link is needed to develop targeted policies and interventions in this area.

## Supporting information

**S1 Table. Cross-sectional and prospective associations between perceived discrimination and health and wellbeing outcomes in the sample without complete body mass index data.** (DOCX)

**S2 Table. Characteristics of complete cases and those lost to follow-up at wave 5 of the English Longitudinal Study of Ageing (2010/11).** (DOCX)

**S3 Table. Cross-sectional and prospective associations between perceived discrimination and health and wellbeing outcomes in those who provided follow-up data.** (DOCX)

**S4 Table. Cross-sectional and prospective associations between perceived discrimination and health and wellbeing outcomes (imputed).** (DOCX)

**S5 Table. Sensitivity analysis: Perceived discrimination measure excluding each discriminatory behaviour in turn.** (DOCX)

## Author Contributions

**Conceptualization:** Ruth A. Hackett.

**Data curation:** Ruth A. Hackett, Sarah E. Jackson.

**Formal analysis:** Ruth A. Hackett.

**Funding acquisition:** Ruth A. Hackett.

**Writing – original draft:** Ruth A. Hackett.

**Writing – review & editing:** Ruth A. Hackett, Myra S. Hunter, Sarah E. Jackson.

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
