## [Decision Letter · Decision Letter 0]

24 Nov 2023

PONE-D-23-27007The relationship between sex discrimination and wellbeing in middle-aged and older womenPLOS ONE

Dear Dr. Hackett, Thank you for submitting your manuscript to PLOS ONE. After careful consideration by a Reviewer and an Academic Editor, all of the critiques by the Reviewer must be addressed in detail in a revision to determine publication status. If you are prepared to undertake the work required, I would be pleased to reconsider my decision, but revision of the original submission without directly addressing the critiques of the Reviewer does not guarantee acceptance for publication in PLOS ONE. If the authors do not feel that the queries can be addressed, please consider submitting to another publication medium. A revised submission will be sent out for re-review. The authors are urged to have the manuscript given a hard copyedit for syntax and grammar.

We look forward to receiving your revised manuscript.

Kind regards,

Stephen D. Ginsberg, Ph.D.

Section Editor

PLOS ONE

Journal Requirements:

Reviewers' comments:

Reviewer's Responses to Questions

**Comments to the Author**

1. Is the manuscript technically sound, and do the data support the conclusions?

Reviewer #1: Yes

2. Has the statistical analysis been performed appropriately and rigorously? 

Reviewer #1: Yes

3. Have the authors made all data underlying the findings in their manuscript fully available?

Reviewer #1: Yes

4. Is the manuscript presented in an intelligible fashion and written in standard English?

Reviewer #1: Yes

5. Review Comments to the Author

Reviewer #1: In this work, the authors show that in a large sample of middle and older women, those who perceive discrimination experience worse mental health and life satisfaction then those who do not. Furthermore, they show that the perception causes the change in health with longitudinal data.

This paper was clear and easy to follow. There is much to like, including a large sample, and two waves of data. Nonetheless, I see remove for improvement, and I hope these comments are useful for the authors.

Researchers have documented that there is variance in the extent to which members of low-status groups acknowledge discrimination as a problem that their group must face (Seller & Shelton, 2003; Stephens & Levine, 2011), including a long tradition of studying women’s tendency to deny personal discrimination (considering Faye Crosby’s work in the 1980s). Furthermore, researchers have shown that people are motivated to minimize group-based discrimination. For example, Napier and colleagues (2020) and Bahamodes and colleagues (2019) have shown that women who minimization discrimination experience better mental and physical health, likely because they are motivated to see the world as fair. Using longitudinal data, Bahamodes and colleagues (2021) have shown that the motivation to see fairness precedes (and reduces) perceptions of discrimination among women. While this work is not capable of addressing these concerns, it is worth mentioning in the discussion section that the tendency to minimize is prevalent and can be motivated.

How is discrimination item scored? Do they get one point for each type of discrimination they said yes to and attributed to sex? Is the range 0-5? Please add a line akin to what is included in the measure of symptomology “The overall score ranged from 0-8, with higher values indicating greater symptomatology”.

Alternatively, it seems that most of the people who report any discrimination report a single instance. Do you have sufficient variance to consider a 0-5 scale (if you indeed did), or would it be more appropriate use a binary variable with 0 (no discrimination) and 1 (at least one event).

Why do you include CI when reporting what % of the sex discrimination events were each of the five scenarios? This is not an estimated number but a reported number (e.g., of those reports of sex discrimination, 82.3% reported being treated with less curtesy or respect…. There is nothing to estimate).

Would be worth directly reporting that just over 9% of your sample is nonwhite.

Finally, I was surprised to see the authors refer to the phenomenon as “sex discrimination”, versus the “gender discrimination”. Perhaps this is field specific, but in my own, authors prefer the use of gender discrimination. We can consider how it would sound in other contexts, for example, sex bias or sex inequality doesn’t quite sound right, and also is not entirely clear, whereas gender bias or gender inequality is clear.

6. PLOS authors have the option to publish the peer review history of their article (what does this mean?). If published, this will include your full peer review and any attached files.

Reviewer #1: No

---

## [Author Response · Author response to Decision Letter 0]

26 Dec 2023

Reviewer #1: In this work, the authors show that in a large sample of middle and older women, those who perceive discrimination experience worse mental health and life satisfaction then those who do not. Furthermore, they show that the perception causes the change in health with longitudinal data. This paper was clear and easy to follow. There is much to like, including a large sample, and two waves of data. Nonetheless, I see remove for improvement, and I hope these comments are useful for the authors.

1. Researchers have documented that there is variance in the extent to which members of low-status groups acknowledge discrimination as a problem that their group must face (Seller & Shelton, 2003; Stephens & Levine, 2011), including a long tradition of studying women’s tendency to deny personal discrimination (considering Faye Crosby’s work in the 1980s). Furthermore, researchers have shown that people are motivated to minimize group-based discrimination. For example, Napier and colleagues (2020) and Bahamodes and colleagues (2019) have shown that women who minimization discrimination experience better mental and physical health, likely because they are motivated to see the world as fair. Using longitudinal data, Bahamodes and colleagues (2021) have shown that the motivation to see fairness precedes (and reduces) perceptions of discrimination among women. While this work is not capable of addressing these concerns, it is worth mentioning in the discussion section that the tendency to minimize is prevalent and can be motivated.

Thank you for taking the time to review this paper and for this very helpful comment. In response we have added the following lines to the Discussion section (page 18):

“Only 9.2% of our sample reported gender discrimination. However, there is evidence that the tendency to minimize or deny personal discrimination is prevalent among women (Stephens & Levine, 2011). This has an impact on mental wellbeing, with evidence from large cohort studies suggesting that denial of gender discrimination is linked with greater mental wellbeing (Napier et al., 2020; Bahamondes et al., 2019). This denial of discrimination is suggested to be motivated by a desire to see the world as fair (known as system-justifying beliefs) and this may be beneficial for mental wellbeing (Napier et al., 2020; Bahamondes et al., 2019; Bahamondes et a., 2021). We were unable to investigate whether system-justifying beliefs influenced the reporting of gender discrimination and in turn the links between gender discrimination and mental wellbeing in this study due to a lack of data availability. This represents an important avenue for future work”.

We have also added the Stephens & Levine, Napier et al., 2020; Bahamondes et al., 2019 and Bahamondes et a., 2021 papers to our reference list. 

2. How is discrimination item scored? Do they get one point for each type of discrimination they said yes to and attributed to sex? Is the range 0-5? Please add a line akin to what is included in the measure of symptomology “The overall score ranged from 0-8, with higher values indicating greater symptomatology”. Alternatively, it seems that most of the people who report any discrimination report a single instance. Do you have sufficient variance to consider a 0-5 scale (if you indeed did), or would it be more appropriate use a binary variable with 0 (no discrimination) and 1 (at least one event).

We are sorry this was unclear in our initial submission. Participants were asked about the frequency of encounters with 5 discriminatory situations. The response options for each of the 5 items were on 6-point scale from ‘never’ to ‘almost every day’. It was more appropriate for us to derive a binary variable for discrimination as the data were skewed. This is in line with previous ELSA work with this measure (see references 24,31-35). 

To clarify this in the text we now state in the Methods section (page 7):

“Response options were on a 6‐point scale ranging from ‘never’ to ‘almost every day’. As the data were skewed, with most women ‘never’ reporting discrimination, we dichotomised responses to indicate whether or not they perceived discrimination in the past year (a few times or more a year vs less than once a year or never), with the exception of the fifth item which was dichotomised to indicate whether or not respondents had ever experienced discrimination from doctors or hospitals (never vs all other options) as most individuals never reported discrimination in this setting. In line with previous work in ELSA, responses were combined to create an overall discrimination binary score (yes/no) if participants reported any of these experiences(24,31–35).

3. Why do you include CI when reporting what % of the sex discrimination events were each of the five scenarios? This is not an estimated number but a reported number (e.g., of those reports of sex discrimination, 82.3% reported being treated with less curtesy or respect…. There is nothing to estimate).

The reviewer is correct 82.3% is a reported not an estimated number. We initially put confidence intervals around the reported discrimination using a one sample t-test to estimate where the prevalence might lie in the population. We are sorry this caused confusion. Therefore, in response we have now removed these confidence intervals from the paper (page 10). 

4. Would be worth directly reporting that just over 9% of your sample is nonwhite.

Thank you for this suggestion. In response to this comment, we now report the prevalence of ethnic minority (non-white) participants in the in text in the Results section (page 10):

“1.8% (n=55) of the sample reported being from an ethnic minority group”. 

Please note the prevalence of ethnic minority (non-white) participants is less than 9%, as ELSA is a very predominately white sample. We note this limitation in the discussion section (page 19) where we say:

“Our sample was largely of white ethnicity. Therefore, our findings may not generalise to ethnic minority groups”. 

5. Finally, I was surprised to see the authors refer to the phenomenon as “sex discrimination”, versus the “gender discrimination”. Perhaps this is field specific, but in my own, authors prefer the use of gender discrimination. We can consider how it would sound in other contexts, for example, sex bias or sex inequality doesn’t quite sound right, and also is not entirely clear, whereas gender bias or gender inequality is clear.

Thank you for this comment. We used the term “sex” in our initial submission as “sex” rather than “gender” was used in the ELSA questionnaire as a possible response option to the question asking participants what characteristic they attributed the discrimination to. However, we appreciate and recognise the reviewers point, so in response to this comment we have changed the term “sex discrimination” to “gender discrimination” throughout the paper. We leave the term “sex” in the Methods section where we talk about the list of options participants could attribute their discrimination to (“with a choice from a list of options including age, race, sex, sexual orientation, and weight).

---

## [Decision Letter · Decision Letter 1]

9 Feb 2024

The relationship between gender discrimination and wellbeing in middle-aged and older women

PONE-D-23-27007R1

Dear Dr. Hackett,

We’re pleased to inform you that your manuscript has been judged scientifically suitable for publication and will be formally accepted for publication once it meets all outstanding technical requirements.

Kind regards,

Stephen D. Ginsberg, Ph.D.

Section Editor

PLOS ONE

**Comments to the Author**

1. If the authors have adequately addressed your comments raised in a previous round of review and you feel that this manuscript is now acceptable for publication, you may indicate that here to bypass the “Comments to the Author” section, enter your conflict of interest statement in the “Confidential to Editor” section, and submit your "Accept" recommendation.

Reviewer #1: All comments have been addressed

2. Is the manuscript technically sound, and do the data support the conclusions?

Reviewer #1: Yes

3. Has the statistical analysis been performed appropriately and rigorously? 

Reviewer #1: Yes

4. Have the authors made all data underlying the findings in their manuscript fully available?

Reviewer #1: Yes

5. Is the manuscript presented in an intelligible fashion and written in standard English?

Reviewer #1: Yes

6. Review Comments to the Author

Reviewer #1: All comments have been addressed. Thank you for the opportunity to review this paper. best of luck with your continued line of research.

7. PLOS authors have the option to publish the peer review history of their article (what does this mean?). If published, this will include your full peer review and any attached files.

Reviewer #1: **Yes: **Alexandra Suppes

---

## [Editor Report · Acceptance letter]

27 Feb 2024

PONE-D-23-27007R1 

PLOS ONE

Dear Dr. Hackett, 

I'm pleased to inform you that your manuscript has been deemed suitable for publication in PLOS ONE. Congratulations! Your manuscript is now being handed over to our production team.

Kind regards, 

on behalf of

Dr. Stephen D. Ginsberg 

Section Editor

PLOS ONE